# Ozone ultrafine bubble water exhibits bactericidal activity against pathogenic bacteria in the oral cavity and upper airway and disinfects contaminated healthcare equipment

Fumio Takizawa[1,2], Hisanori Domon[1,3], Takumi Hiyoshi[1,2,3], Hikaru Tamura[1,2,3], Kana Shimizu[1], Tomoki Maekawa[1,2,3], Koichi Tabeta[2], Akiomi Ushida[4], Yutaka Terao[1,3]*

1 Division of Microbiology and Infectious Diseases, Niigata University Graduate School of Medical and Dental Sciences, Niigata, Japan, 2 Division of Periodontology, Niigata University Graduate School of Medical and Dental Sciences, Niigata, Japan, 3 Center for Advanced Oral Science, Niigata University Graduate School of Medical and Dental Sciences, Niigata, Japan, 4 Institute of Science and Technology, Niigata University, Niigata, Japan

* terao@dent.niigata-u.ac.jp

**Data Availability Statement:** All relevant data are within the manuscript and its Supporting Information files.

## Abstract

Ozone is strong oxidizing agent that is applied in aqueous form for sanitation. However, ozonated water is unstable and has a short half-life. Ultrafine bubble technology is promising to overcome these issues. Ultrafine bubble is nanoscale bubble and can exist in water for a considerable duration of time. This study aims to investigate the application of ozone ultra-fine bubble water (OUFBW) as a disinfectant. We produced an OUFBW generator which generates OUFBW containing 4–6 ppm of ozone. Thereafter, we examined the bactericidal activity of the OUFBW against various pathogenic bacteria in oral cavity and upper airway, including antibiotic-susceptible and antibiotic-resistant *Streptococcus pneumoniae*, *Pseudomonas aeruginosa*, *Streptococcus mutans*, *Streptococcus sobrinus*, *Fusobacterium nucleatum*, *Prevotella intermedia*, and *Porphyromonas gingivalis*. Exposure of planktonic culture of these bacterial species to OUFBW reduced viable bacteria by > 99% within 30s. Additionally, OUFBW exerted bactericidal activity against *S. pneumoniae* and *P. aeruginosa* adhered to toothbrush and gauze, respectively. We also observed disruption of bacterial cell wall of *S. pneumoniae* exposed to OUFBW by transmission electron microscope. Additionally, OUFB did not show any significant cytotoxicity toward the human gingival epithelial cell line Ca9-22. These results suggest that OUFBW exhibits bactericidal activity against broad spectrum of bacteria and has low toxicity towards human cells.

## Introduction

Healthcare-associated infection (HAI) is major threat for hospitalized patients and nursing home residents [1, 2]. Numerous studies indicate that healthcare workers' hands and

**Funding:** This study was funded by Terumo Life Science Foundation (Grant No. 22-III1003 to YT). This work was supported by the Japan Society for the Promotion of Science (JSPS) KAKENHI (grants JP20H03858 and JP22K19614 to YT; JP20K09903 to HD; JP22H03267 to TM; JP19H03829 to KT) and JST, the establishment of University fellowships towards the creation of science technology innovation, Grant Number JPMJFS2114 to FT. The funders had no role in study design, data collection and analysis, decision to publish, or preparation of the manuscript. The specific roles of these authors are articulated in the 'author contributions' section.

**Competing interests:** The authors have declared that no competing interests exist.

healthcare equipment are the source of propagation of micro-organisms to patients [3, 4]. Adequate hand hygiene and cleaning of the equipments can prevent HAI; however, large numbers of patients suffer from HAI annually, and resulting in significant financial and individual costs [3]. In addition, there has been an alarming rise in HAI by multi-drug-resistant bacteria, and it causes a major threat around the world [5, 6]. Therefore, it is necessary to explore a new disinfectant which is safe, low-cost and effective against various pathogens including drug-resistant bacteria.

Ozone is a strong oxidizing agent and exerts antimicrobial activity against bacteria, fungi, protozoa, and viruses and does not induce microbial resistance [7–9]. Therefore, ozone was originally noted in the fields of water purification [10] and food industry [11]. On the other hand, high concentration of ozone may be harmful to humans, for example, exposure to 0.08 ppm ozone is associated with increased airway inflammation after 18 hours of exposure [12, 13].

Previous studies have indicated that ozonated water is shown to be a powerful antimicrobial agent like ozone gas and is less harmful than ozone gas, because ozonic volatilization from the surface of ozonated water is very low [14]. However, since ozone in aqueous solution rapidly degrades to oxygen [15], ozonated water has a half-life of only 1 hour or less and must be used within 5 to 10 min after production to exert its bactericidal activity. In order to overcome these issues, ultrafine bubble (UFB) technology is gaining attention.

UFB is less than 200 nm in diameter which can exist in water for a considerable duration of time [16]. The stability of UFB is attributed to the electrically charged liquid-gas interface, which creates repulsion forces that prevent the bubble coalescence, and the high dissolved gas concentration in the water, which keeps a small concentration gradient between the interface and the bulk liquid [17]. In this regard, ozone ultrafine bubble water (OUFBW) is expected to use as new bactericidal agent, and several researches of OUFBW have been conducted [18–20].

In this study, we produced OUFBW generator which can generate OUFBW containing high concentration (4–6 ppm) of ozone; then we analyzed the characteristic and bactericidal activity of OUFBW application as disinfectant. We also investigated bactericidal mechanism and cytotoxicity of OUFBW against human cells.

## Materials and methods

### Production and characteristics of OUFBW

Ozone gas was generated by dielectric barrier discharge ozone generator (10 g/h) (Futech-Niigata LLC, Niigata, Japan) with 90% oxygen gas provided by oxygen concentrator (flow rate of 1 L/min) (BMC Medical Co., Ltd, Beijing, China). OUFBW was generated by micro blender (Futech-Niigata LLC) and re-circulated in the polyvinyl chloride water tank. Generation of UFBs needed more than 0.2 MPa of pump pressure, however, increasing pump pressure caused an increase in water temperature, leading to decreased ozone concentration of OUFBW. To prevent this, as shown in Fig 1A, a cooling tank was assembled to keep water temperature 10˚C or less, which enabled us to generate OUFBW containing high concentration (4–6 ppm) of ozone. Thereafter, the changes of ozone concentration in OUFBW which stored in light-blocking bottle (Corning, NY, USA) was measured by Digital Pack Test Ozone (Kyoritsu Chemical Check Lab, Tokyo, Japan) every 2 hours [21]. The diameter of OUFBW was measured by Nano Particle Size Analyzer (SALD-7500nano; Shimadzu, Kyoto, Japan).

### Bacterial culture and reagents

All Gram-positive bacteria used in this study were cultured at 37˚C under aerobic conditions. *Streptococcus mutans* strain MT8148 and *Streptococcus sobrinus* strain MT10186 were grown

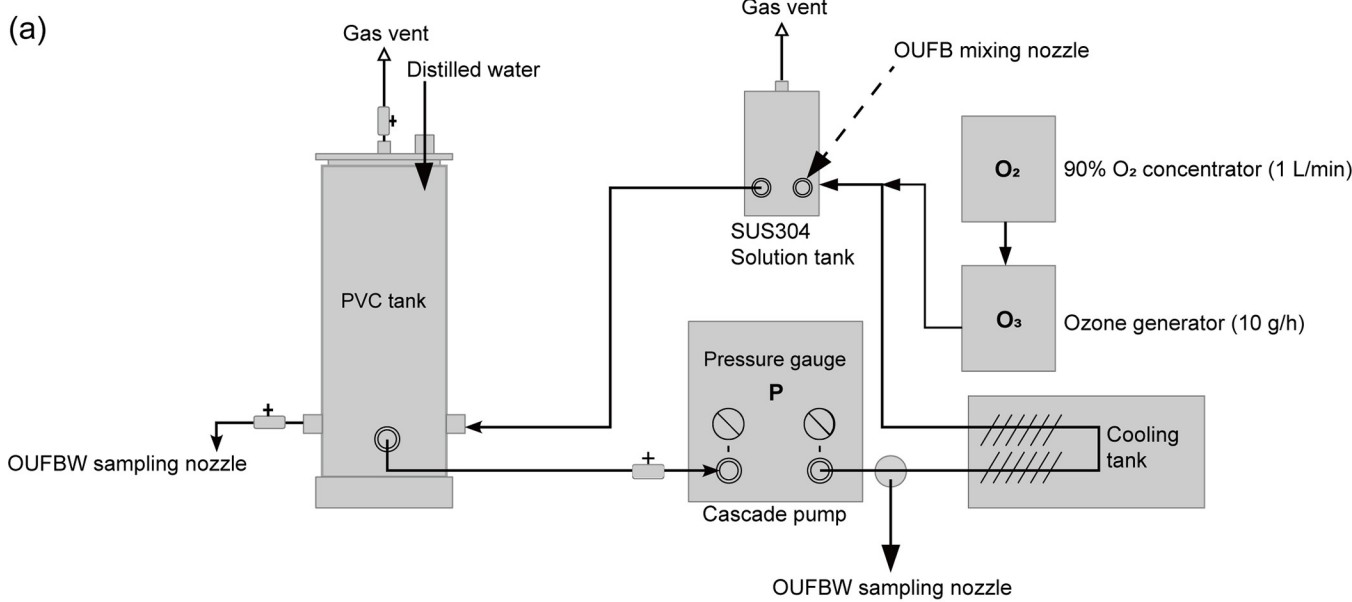

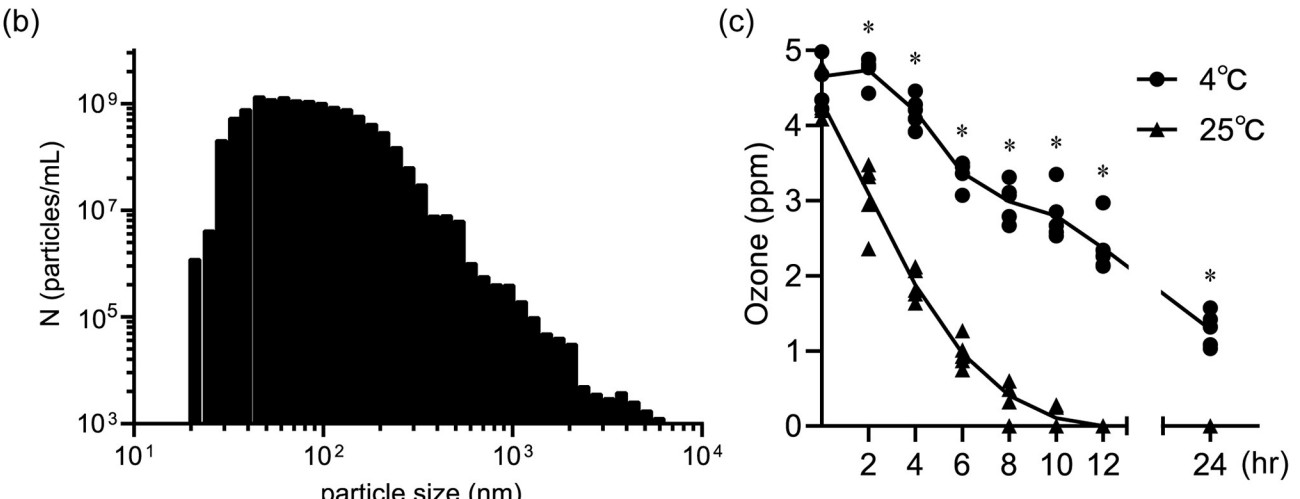

**Fig 1. Production of ozone ultrafine bubble water (OUFBW).** (a) Schematic diagram of OUFBW generator. (b) Particle size distribution of OUFBs. (c) Changes of ozone concentration in OUFBW stored at 4°C or 25°C. Data are presented as the mean ± SD of quintuplicate experiments and were evaluated using two-way repeated measures analysis of variance. Statistically significant as compared to 4°C group, *$P < 0.05$.

in brain heart infusion broth (Becton Dickinson, Franklin Lakes, NJ, USA) for 24 hr [22]. Macrolide-resistant *Streptococcus pneumoniae* strain NU4471 (azithromycin minimum inhibitory concentration $\geq$ 1 mg/mL) isolated from patients with respiratory tract infections [23], multidrug-resistant *S. pneumoniae* strain KM256 (penicillin G, ceftriaxone, azithromycin, and levofloxacin minimum inhibitory concentrations of $>$ 8, 8, 4, and $>$ 8 μg/mL, respectively) isolated from children with acute otitis media [24], and antibiotic-susceptible *S. pneumoniae* strain D39 were grown in tryptic soy broth (TSB; Becton Dickinson) for 12 hr. All Gram-negative obligate anaerobes used in this study were cultured at 37°C in an anaerobic jar (Mitsubishi Gas Chemical, Tokyo, Japan) using an AnaeroPack anaerobic cultivation system (Mitsubishi

Gas Chemical) at 37˚C [22, 25]. *Porphyromonas gingivalis* strain ATCC 33277, *Prevotella intermedia* strain ATCC 25611, and *Fusobacterium nucleatum* strain ATCC 25586 were grown in modified Gifu anaerobic medium broth (Nissui, Tokyo, Japan) under anaerobic conditions for 48 hr [26]. *Pseudomonas aeruginosa* strain RIMD 1603003 was grown in tryptic soy broth (Becton Dickinson) under aerobic conditions for 12 hr. The overnight cultures were then inoculated into the relevant culture medium and allow to grow until they reached exponential growth phase (optional density (OD) at 600 nm of 0.1–0.5). Bacteria were subsequently used for bactericidal activity assays.

## Measurement of bactericidal activity of OUFBW

The bactericidal activity of OUFBW against planktonic cells was analyzed by standard plating methods [27]. The bacterial cells were exposed to OUFBW by two different methods. First, the 1 µL bacterial culture with $OD_{600}$ values of 0.1–0.5 was added to 1 mL of OUFBW containing high concentration (4–6 ppm) of ozone and incubated for 10–300 s. Second, 1 µL bacterial culture was added to 1 mL of OUFBW containing various concentration (0.25–6 ppm) of ozone for 1 min. To prepare OUFBW containing various ozone concentrations, OUFBW containing 4–6 ppm ozone was serially diluted with distilled water. Thereafter, OUFBW-exposed bacterial cultures were diluted in distilled water, after which *S. mutans* and *S. sobrinus* were cultured on Mitis-Salivarius agar plates (Becton Dickinson), *P. aeruginosa* on TSB agar plates (Becton Dickinson), *S. pneumoniae* strains, *P. gingivalis*, *P. intermedia*, and *F. nucleatum* on trypticase soy agar II 5% sheep blood plates (Becton Dickinson). The agar plates were then incubated under aerobic (Gram-positive bacteria and *P. aeruginosa*) or anaerobic (other Gram-negative bacteria) conditions at 37˚C for 2–7 days.

## Transmission Electron Microscope (TEM) observation

After 1 mL *S. pneumoniae* NU4471 bacterial culture ($OD_{600}$ = 0.1) was added to 49 ml of ozone ultrafine bubble water (1 ppm) or distilled water and incubated for 1 min, the bacterial cells were harvested by centrifugation at $10,000 \times g$ for 15 min and fixed in 2% glutaraldehyde in phosphate buffer. Subsequently, the samples were treated with potassium permanganate, and post-fixed in 2% osmium tetra oxide for 2 hours at 4˚C. The specimens were dehydrated in a graded ethanol and embedded in the epoxy resin. Ultrathin sections (80–90 nm) were obtained by ultramicrotome technique. Ultrathin sections were stained with uranyl acetate for 15 min, and lead staining solution for 5 min. The samples were submitted to TEM (JEOL 1400Flash, Akishima, Tokyo, Japan) observation at 100 kV. The sample preparation and TEM observation were conducted at Filgen, Inc (Nagoya, Japan) [28].

## Sterilization of medical and dental equipment using OUFBW

First, commercially available toothbrushes were immersed in bacterial cultures of *S. pneumoniae* NU4471 ($OD_{600}$ = 0.1) and naturally dried at room temperature. The contaminated toothbrushes were then immersed in 100 mL of OUFBW containing approximately 5 ppm ozone, 0.1% povidone iodine, or distilled water for 5 min. After exposure, the toothbrushes were transferred into 7 mL distilled water, and bacteria attached to the toothbrushes was harvested by ultrasonication (43 kHz, 5 min). Ultrasonicated solution was inoculated on to trypticase soy agar II 5% sheep blood agar. The agar plates were then incubated under aerobic conditions at 37˚C for 2 days.

Next, sterilized gauzes (5 cm × 5 cm) were moistened with 250 µL bacterial cultures of *P. aeruginosa* RIMD 1603003 ($OD_{600}$ = 0.1) and naturally dried at room temperature. The contaminated gauzes were immersed in 100 mL or 500 mL of OUFBW containing approximately

5 ppm ozone, 0.1% NaClO, or distilled water for 5 min. After exposure, the gauzes were transferred into 5 mL distilled water, and bacteria attached to the gauzes was harvested by ultrasonication (43 kHz, 5 min). Ultrasonicated solution was diluted in distilled water and inoculated on to TSB agar plate. The agar plates were then incubated under aerobic conditions at 37˚C for 2 days.

### Cytotoxicity of OUFB against human gingival epithelial cell line

The human gingival epithelial cell line Ca9-22 (RIKEN Cell Bank, Ibaraki, Japan) was grown in minimum essential medium (Thermo Fisher Scientific, Waltham, MA, USA) supplemented with 10% fetal bovine serum (Japan Bio Serum, Hiroshima, Japan), 100 U/mL penicillin, and 100 μg/mL streptomycin (Wako Pure Chemical Industries, Osaka, Japan) at 37˚C in an atmosphere of 95% air and 5% $CO_2$. These cells were seeded at a density of $1 \times 10^5$ cells/100 μL in 96-well plates 24 hr prior to treatment. For cytotoxicity assay, we prepared OUFB- phosphate-buffered saline (PBS) by diluting OUFBW with $10 \times$ PBS (NACALAI TESQUE INC, Kyoto, Japan). Thereafter cells were exposed to 100 μL of OUFB-PBS containing various concentration (1–5 ppm) of ozone, 0.1% povidone iodine, 0.2% chlorhexidine, 0.1% NaClO, or 0.5% Triton X-100 for 1 or 30 min. After treatment, the cells were carefully washed with PBS to remove any residual activities of reagents. Cellular viability was assessed using the MTT [3-(4,5-dimethythiazol-2-yl)-2,5-diphenyl tetrazolium bromide] assay. The OD of the colored solution was quantified spectrophotometrically at 571 nm using a microplate reader (Multiskan FC, Thermo, Waltham, MA, USA).

### Statistical analysis

All data were evaluated statistically by analysis of variance with a Dunnett's multiple comparisons test using GraphPad Prism Software (version 9; GraphPad Software, La Jolla CA, USA). $P < 0.05$ was considered to denote statistical significance.

## Results

### Ozone ultrafine bubbles were remained stable at 4˚C

We first examined the characteristics of the OUFBW. Fig 1B shows particle diameter distribution of the OUFBs. The average diameter of OUFBs was 70 ± 225 nm and the most frequent diameter of OUFBs was 45 nm. We next examined storage temperature for OUFBW. When OUFBW was stored at 25˚C, ozone concentration decreased to 0 ppm after 12 hr, suggesting the OUFBs is unstable at room temperature. In contrast, storage at 4˚C caused OUFBs stable, and the ozone concentration was more than 1 ppm after 24 hr (Fig 1C).

### OUFBW exerts bactericidal effects against various bacterial species

We conducted two experiments to investigate bactericidal effects of OUFBW against various bacterial species, such as respiratory pathogen (antibiotic-resistant and antibiotic-susceptible *S. pneumoniae*), opportunistic bacterium (*P. aeruginosa*), cariogenic bacteria (*S. mutans* and *S. sobrinus*), and periodontopathic bacteria (*P. gingivalis*, *P. intermedia*, and *F. nucleatum*). First, various bacteria were exposed to OUFBW containing high concentration (4–5 ppm) of ozone for 10–300 s. Colonies were undetected in all bacterial species after > 30 s of exposure to OUFBW (Table 1). We next exposed the bacteria to OUFBW containing various concentration (0.25–6 ppm) of ozone for 1 min. Fig 2 shows that exposure to approximately ≥ 1 ppm OUFBW resulted in a > 99% decrease in the viability of all bacterial species including

**Table 1. The number of colony-forming units of various bacterial species exposed to ozone ultrafine bubble water.**

| Bacterial species (Bacterial strains) | CFU/mL ± 1SD | | | | | |
|---|---|---|---|---|---|---|
| | **Baseline** | **10 s** | **30 s** | **60 s** | **150 s** | **300 s** |
| *S. pneumoniae* (strain D39) | $(1.3 \pm 0.5) \times 10^8$ | ND | ND | ND | ND | ND |
| *S. pneumoniae* (strain NU4471) | $(4.2 \pm 0.7) \times 10^8$ | ND | ND | ND | ND | ND |
| *S. pneumoniae* (strain KM256) | $(3.2 \pm 0.1) \times 10^7$ | ND | ND | ND | ND | ND |
| *P. aeruginosa* (strain RIMD 1603003) | $(3.3 \pm 0.9) \times 10^7$ | ND | ND | ND | ND | ND |
| *S. mutans* (strain MT8148) | $(4.9 \pm 1.8) \times 10^8$ | ND | ND | ND | ND | ND |
| *S. sobrinus* (strain MT10186) | $(6.6 \pm 2.1) \times 10^8$ | ND | ND | ND | ND | ND |
| *F. nucleatum* (strain ATCC25586) | $(6.2 \pm 0.6) \times 10^8$ | ND | ND | ND | ND | ND |
| *P. intermedia* (strain ATCC 25611) | $(5.8 \pm 0.9) \times 10^7$ | ND | ND | ND | ND | ND |
| *P. gingivalis* (strain ATCC 33277) | $(4.5 \pm 1.0) \times 10^8$ | $(0.5 \pm 0.3) \times 10^8$ | ND | ND | ND | ND |

The number of colony-forming units (CFU) of various bacterial species was measured after exposing to ozone ultrafine bubble water containing 4–6 ppm of ozone for 10–300 s. ND stands for undetected and indicates below the detection limit ($< 100$ CFU/mL).

antibiotic-resistant *S. pneumoniae* strains. These results indicate that OUFBW exerts bactericidal effect instantly and nonspecifically.

## OUFBW treatment causes morphologic change of bacterial cells

To investigate bactericidal mechanism of OUFBW, we attempted to observe *S. pneumoniae* exposed to OUFBW by TEM, however, when *S. pneumoniae* was exposed to 5 ppm OUFBW, we could not detect bacterial cell pellet by centrifuge. Instead, we used 1 ppm OUFBW, which enabled us to detect pneumococcal cell pellet by centrifuge. TEM analysis showed that OUFBW damaged the cell wall and made crevice between cell wall and cytoplasm (Fig 3).

## OUFBW sterilizes medical and dental equipment

Oral bacteria, including *S. pneumoniae, P. intermedia and F. nucleatum*, are known as a common pathogen that cause aspiration pneumonia [29]. Although oral care can decrease the incidence of aspiration pneumonia [30], previous study indicates that retention and survival of bacteria on toothbrushes after brushing represent a possible cause of re-contamination of the mouth [31]. Thus, we used OUFBW as a disinfectant for toothbrushes contaminated with macrolide-resistant *S. pneumoniae*. Contaminated toothbrushes were immersed in approximately 5 ppm OUFBW or distilled water for 5 min. Exposure to OUFBW resulted in a $> 90\%$ decrease in the bacterial load of *S. pneumoniae* compared to that of the distilled water treatment group (Fig 4A).

Additionally, *P. aeruginosa* is a major cause of nosocomial infection. Previous study suggested that environment, such as bed sheet and cloth, could be a main reservoir for *P. aeruginosa* in hospital [32]. Therefore, we next used OUFBW as a disinfectant for cotton gauzes contaminated with *P. aeruginosa*. In this study, the gauzes were likened to cloth product in hospital and nursing home. Contaminated gauzes were immersed in approximately 5 ppm OUFBW or distilled water for 5 min. Exposure to OUFBW resulted in a $> 90\%$ decrease in the bacterial load of *P. aeruginosa* compared to that of the distilled water treatment group (Fig 4B).

## OUFB has minimum cytotoxic effect toward human cell line Ca9-22

To use OUFB as a disinfectant for medical and dental equipment, we analyzed its cytotoxicity towards the human gingival epithelial cell line Ca9-22. Exposure to OUFB-PBS (1–5 ppm of

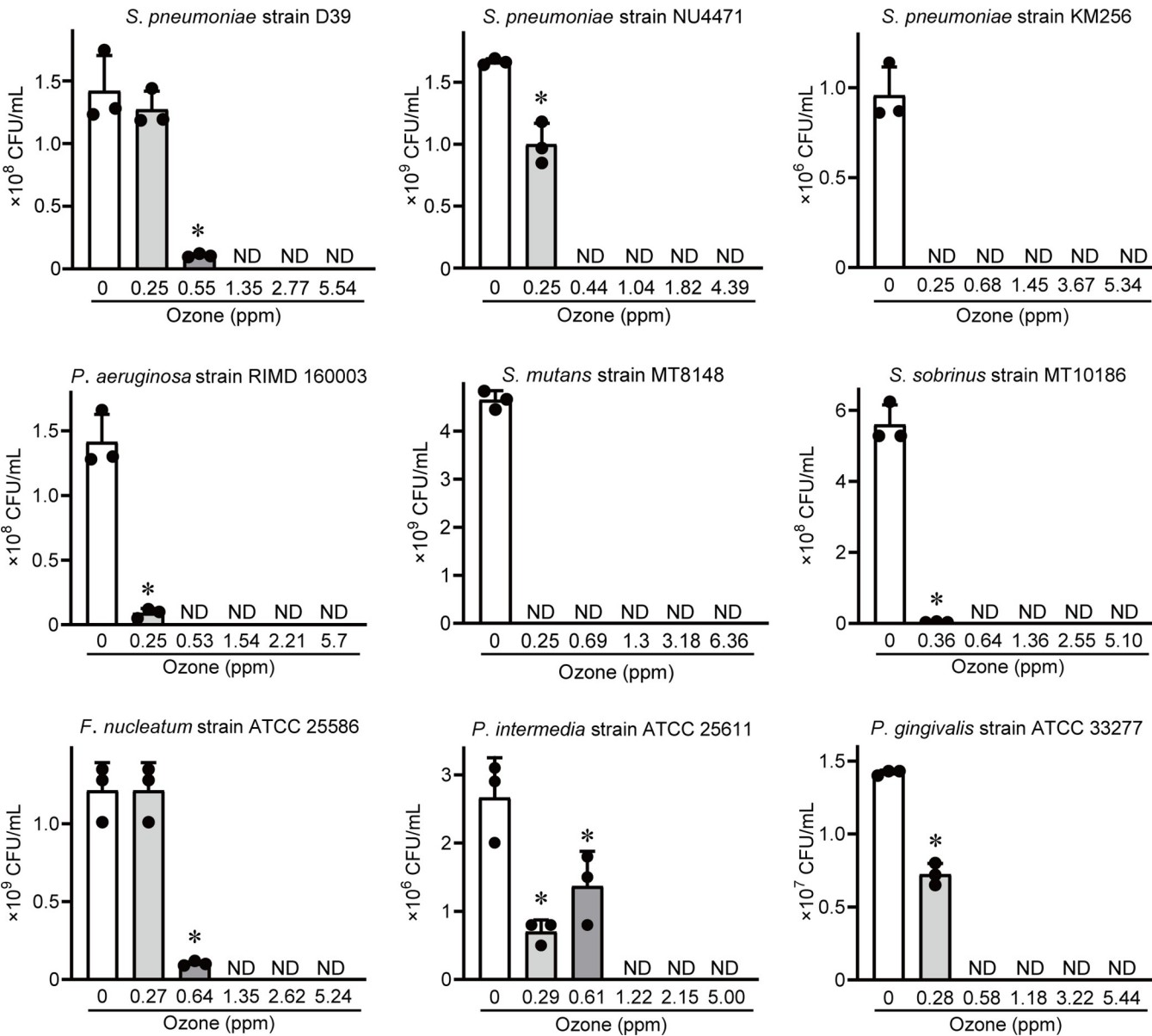

**Fig 2. Bactericidal effects of ozone ultrafine bubble water (OUFBW) against various bacterial species.** Various bacterial species were exposed to 0.25–6 ppm OUFBW for 1 min. Data are presented as the mean ± SD of triplicate experiments and were evaluated using analysis of variance with a Dunnett's multiple-comparisons test. *$P < 0.05$ compared to control group. ND stands for undetected and indicates below the detection limit (< 100 CFU/mL).

ozone concentration) for 1–30 min did not significantly decrease the viability of Ca9-22 cells; however, the viability of the cells treated with 0.1% povidone iodine, 0.2% chlorhexidine, or 0.1% NaClO decreased by > 99% (Fig 5). These results indicated that OUFB has low toxicity to mammalian cells compared to other disinfectants.

## OUFB-PBS exerts bactericidal effects against Gram-positive and negative bacteria

We conducted additional experiment to investigate the bactericidal activity of OUFB-PBS. For this experiment, we used *S. pneumoniae* strain D39 as a Gram-positive bacteria and *P.*

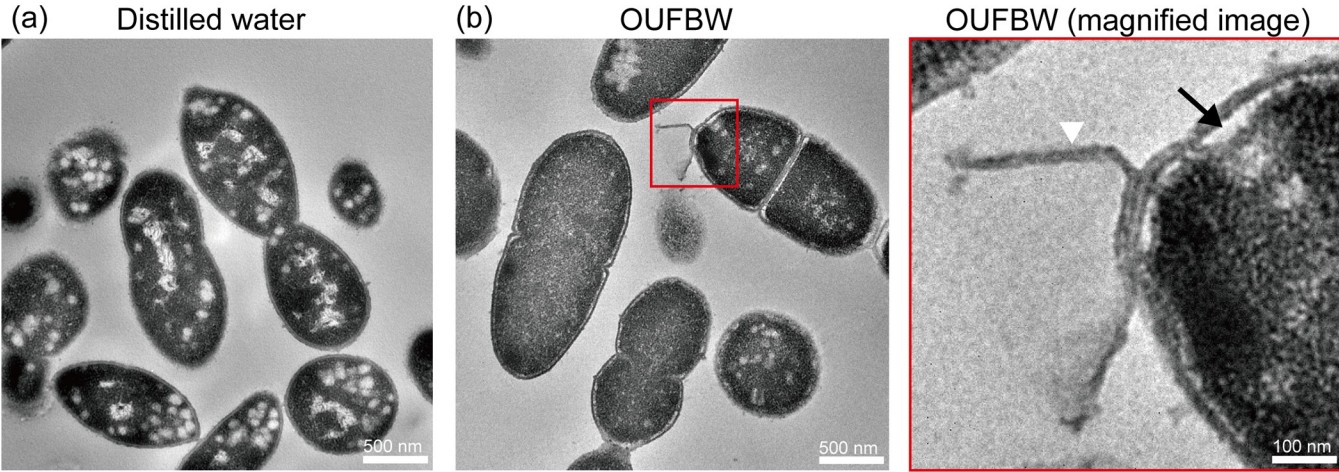

**Fig 3. Transmission electron microscope observation of *S. pneumoniae* exposed to ozone ultrafine bubble water (OUFBW).** Representative transmission electron micrographs of macrolide-resistant *S. pneumoniae* NU4471 after exposure to (a) distilled water or (b) 1 ppm OUFBW. The black arrow indicates the crevice between cell wall and cytoplasm, and the white arrow head indicates the cell wall debris.

*aeruginosa* strain RIMD 1603003 as a Gram-negative bacteria. These bacteria were exposed to OUFB-PBS containing various ozone concentrations (0.25–5 ppm) for 1 min. S1 Fig shows that exposure to $\geq 2$ ppm OUFB-PBS resulted in a $> 99\%$ decrease in *S. pneumoniae*, and exposure to $\geq 1$ ppm OUFB-PBS resulted in a $> 99\%$ decrease in *P. aeruginosa*. These findings suggest that OUFB-PBS also exerts bactericidal effects.

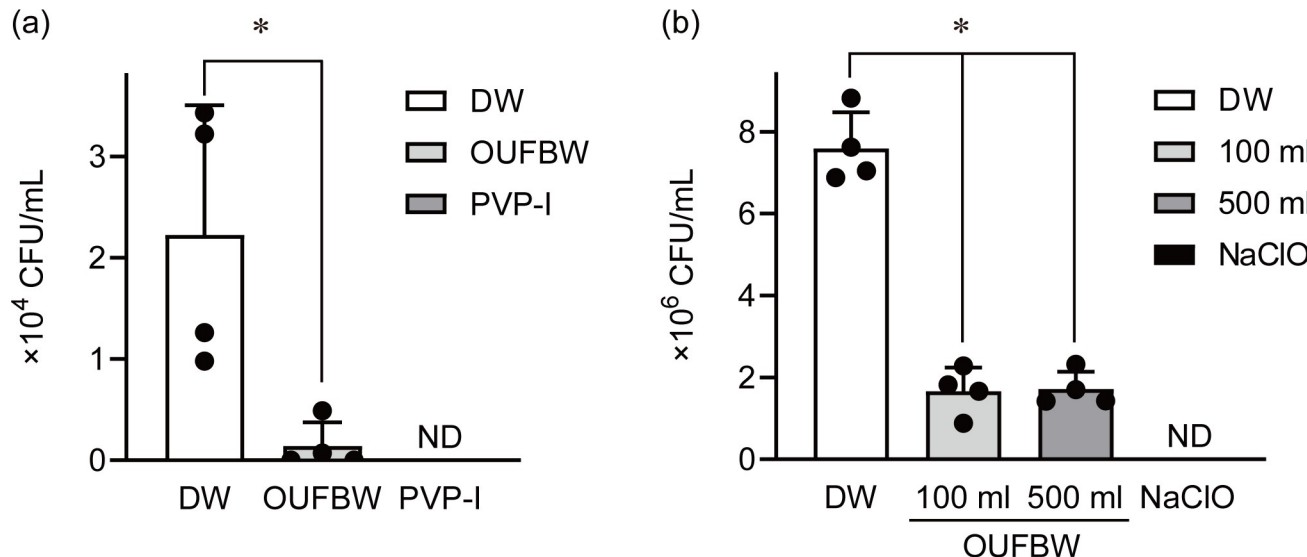

**Fig 4. Bactericidal activity of ozone ultrafine bubble water (OUFBW) against contaminated toothbrushes and gauzes.** (a) The toothbrushes contaminated with *S. pneumoniae* and (b) The gauzes contaminated with *P. aeruginosa* were immersed in distilled water, OUFBW (approximately 5 ppm of ozone concentration), 0.1% povidone iodine or 0.1% NaClO for 5 min. (a, b) Toothbrushes and gauzes were transferred into fresh distilled water (toothbrush: 7 mL, gauze: 5 mL), and then the bacteria attached them were harvested by ultrasonication (43 kHz, 5 min). Bacterial load of *S. pneumoniae* and *P. aeruginosa* were determined by colony count. Data represent the mean ± SD of quadruplicate determinants, and were evaluated using Dunnett's multiple-comparisons test. *$P < 0.05$ versus the control group. ND stands for undetected and indicates below the detection limit (a: $< 70$ CFU/mL, b: $< 5 \times 10^3$ CFU/mL).

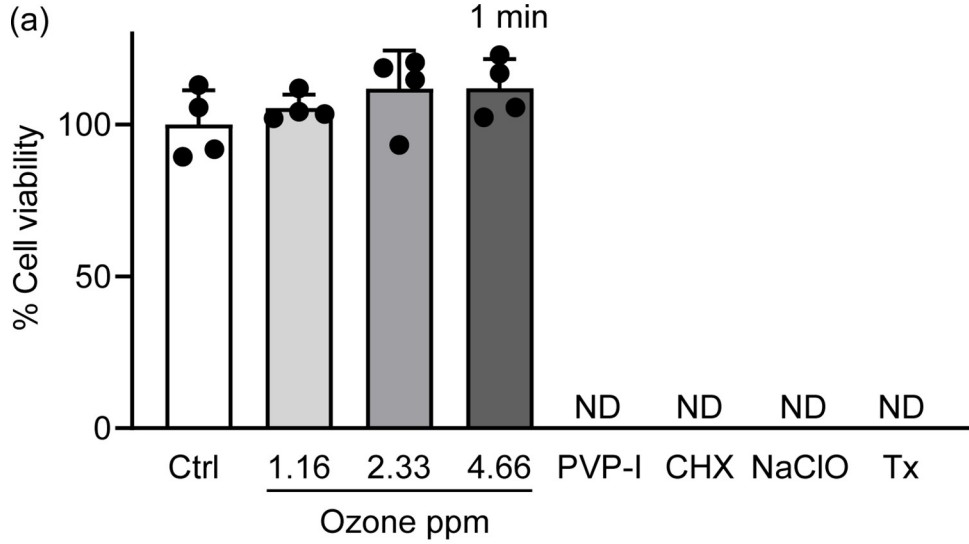

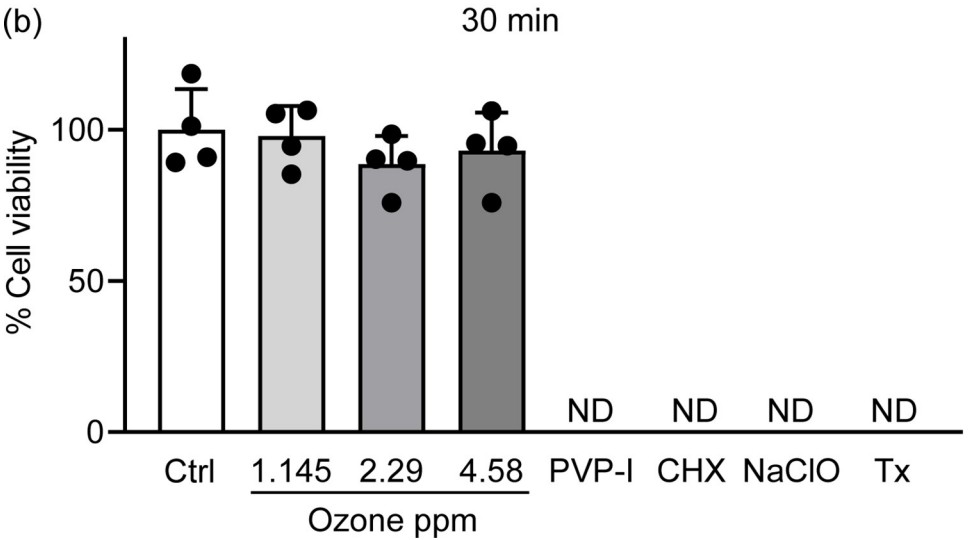

**Fig 5. Cell viability of human gingival epithelial cells after exposure to ozone ultrafine bubble.** Human gingival cell line Ca9-22 were exposed to PBS containing 1–5 ppm ozone ultrafine bubble, 0.1% Povidone iodine, 0.2% Chlorhexidine, 0.1% NaClO or 0.5% Triton X-100 for (a) 1 min or (b) 30 min. MTT (3-(4,5-Dimethyl-2-thiazolyl)-2,5-diphenyl-2H-tetrazolium bromide) assays were performed to determine cell viability. Data represent the mean ± SD of quadruplicate determinants, and were evaluated using Dunnett's multiple-comparisons test. ND stands for undetected and indicates below the detection limit.

## Discussion

HAI is caused by various micro-organisms including drug-resistant bacteria. Since healthcare-associated pathogens can survive for days to weeks on environmental surfaces [33], items frequently touched by healthcare workers, hospitalized patients or nursing home residents are

contaminated by such pathogens. In order to prevent HAI, it is necessary for disinfectants used on a daily basis to have a broad spectrum of antimicrobial activity. In this study, we analyzed the characteristic and bactericidal activity of OUFBW and investigated its practicality as a disinfectant.

It has been reported that OUFBW exerts bactericidal activity against various micro-organisms, such as, *Escherichia coli*, *P. gingivalis*, *Aggregatibacter actinomycetemcomitans*, and *Candida albicans* [18, 20, 34]. In this study, we found that OUFBW also exhibits bactericidal activity against *S. pneumoniae* including antibiotic-resistant strains, *P. aeruginosa*, *S. mutans*, *S. sobrinus*, *F. nucleatum*, and *P. intermedia*. The daily necessaries such as toothbrush and cloth product used in hospital and nursing home were exposed various pathogenic micro-organisms, and they can become the source of HAI. Besides, it was reported that aspiration of pharyngeal bacteria is the major route of infection in the development of nosocomial pneumonia and dental plaque may constitute reservoir of respiratory pathogens [35, 36]. Clinical study has reported that application of OUFBW as adjunct to ultrasonic debridement reduced the number of oral bacteria (*P. gingivalis* and *Tannerella forsythia*) in subgingival plaque compared to tap water significantly [37]. Hence, we supposed that sterilization of healthcare-associated pathogens and oral pathogenic bacteria by OUFBW could contribute the prevention of HAI. In order to investigate the effectiveness of OUFBW as disinfectant for healthcare equipment and oral cavity, we used above-mentioned pathogenic bacteria and found that OUFBW exerted bactericidal activity against all bacterial species. Therefore, we conclude that OUFBW has a broad spectrum of bactericidal activity. In addition, Fig 2 and S1 Fig showed that there was slight difference between OUFBW and OUFB-PBS on the bactericidal activity. The effects of solvents on ozone and nanobubbles have not been investigated, therefore we supposed it should be addressed in future work.

Previous studies indicated that morphological changes in *S. mutans* and *C. albicans* exposed to ozonated water and OUFBW were observed by scanning electron microscopy, respectively [20, 38]; however, the precise bactericidal mechanism of ozone was not fully understood. In order to investigate the mechanism, we conducted TEM analysis of macrolide-resistant *S. pneumoniae* exposed to OUFBW. We first tried to harvest bacterial cells exposed to OUFBW containing approximately 5 ppm of ozone; however, bacterial cell pellet could not be detected after centrifugation. These findings suggest that exposure to OUFBW containing high concentration of ozone might disrupt bacterial cells completely, and thus bacterial cells could not be harvested by centrifugation. Consequently, we observed bacterial cells exposed to OUFBW containing low concentration of ozone (approximately 1 ppm), and found that bacterial cell wall of macrolide-resistant *S. pneumoniae* was damaged and peeled off. This is the first study to show that ozone causes disruption of bacterial cell wall. We also suggest that the bactericidal activity of OUFBs does not depend on anti-microbial resistant.

Although previous studies have demonstrated that OUFBW containing various concentration of ozone (1.5–11 ppm) exerted bactericidal activity [18, 20, 34], the minimum ozone concentration of OUFBW enough to exert bactericidal activity had been not clear. Therefore, we conducted bactericidal assay with OUFBW containing various concentration (0.25–6 ppm) of ozone, and found that exposure to OUFBW containing more than 1 ppm of ozone for more than 30 s caused > 99% decrease of the viability of all bacterial species used in this study. Besides, Fig 1C showed that ozone concentration of OUFBW stored at 4°C was more than 1 ppm after 24 hr. Therefore, it was suggested that OUFBW we produced can exert bactericidal effect during 24 hours after production. On the other hand, OUFBW containing approximately 5 ppm of ozone was not able to completely sterilize bacteria adhered to toothbrush and gauze. Similarly, Shichiri-Negoro *et al.* reported that OUFBW (6–11 ppm) could not completely remove the biofilms of *C. albicans* formed within 24 hr [20]. These results

suggested OUFBs could not permeate complicated structure such as mesh of gauze, bristles of toothbrush and biofilm. In order to enhance bactericidal activity of OUFBW, it is required that planktonic form of bacterial cells is exposed to OUFB directly. For instance, we supposed that suspending bacterial cell in OUFBW by ultrasonication might improve the bactericidal activity.

Present study showed that OUFBW exerts potent bactericidal activity which is effective against drug-resistant bacteria, and has low toxicity toward human gingival epithelial cells. Besides, we found the detail of the bactericidal mechanism of OUFBW. These results suggested that the use of OUFBW as disinfectant in hospital and nursing home would be promising. In order to applicate OUFBW as a disinfectant, further study is required to explore optimal application method and condition.

## Supporting information

**S1 Fig. Bactericidal effects of OUFB-PBS against *S. pneumoniae* strain D39 and *P. aeruginosa* strain RIMD 1603003.** *S. pneumoniae* strain D39 and *P. aeruginosa* strain RIMD 1603003 were exposed to 0.4–5 ppm OUFBW for 1 min. Data are presented as the mean ± SD of triplicate experiments and were evaluated using analysis of variance with Dunnett's multiple-comparisons test. *$P < 0.05$ compared to the control group. ND stands for undetected and indicates below the detection limit (< 100 CFU/mL).
(TIF)

## Acknowledgments

We thank Dr. Satoru Hirayama, Dr. Toshihito Isono, Dr. Karin Sasagawa, Dr. Rui Saito, Dr. Yoshihito Yasui (Niigata University), Mr. Tadashi Hiwatashi (Futech-Niigata LLC), and Mr. Koichi Seto (IWASE Company Limited, Niigata, Japan) for their technical support. We also acknowledge Filgen, Inc for sample preparation and observation of TEM.

## Author Contributions

**Conceptualization:** Hisanori Domon, Akiomi Ushida, Yutaka Terao.

**Formal analysis:** Fumio Takizawa, Hisanori Domon, Takumi Hiyoshi, Yutaka Terao.

**Funding acquisition:** Fumio Takizawa, Hisanori Domon, Tomoki Maekawa, Koichi Tabeta, Yutaka Terao.

**Investigation:** Fumio Takizawa, Hisanori Domon, Takumi Hiyoshi, Hikaru Tamura, Kana Shimizu, Akiomi Ushida, Yutaka Terao.

**Methodology:** Fumio Takizawa, Hisanori Domon, Takumi Hiyoshi, Hikaru Tamura, Kana Shimizu, Akiomi Ushida, Yutaka Terao.

**Project administration:** Fumio Takizawa, Hisanori Domon, Akiomi Ushida, Yutaka Terao.

**Resources:** Hisanori Domon, Yutaka Terao.

**Supervision:** Hisanori Domon, Takumi Hiyoshi, Hikaru Tamura, Akiomi Ushida, Yutaka Terao.

**Writing – original draft:** Fumio Takizawa.

**Writing – review & editing:** Hisanori Domon, Tomoki Maekawa, Koichi Tabeta, Akiomi Ushida, Yutaka Terao.

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
