## [Decision Letter · Decision Letter 0]

7 Feb 2023

PONE-D-23-01776Ozone ultrafine bubble water exhibits bactericidal activity against pathogenic bacteria in the oral cavity and upper airway and disinfects contaminated healthcare equipmentPLOS ONE

Dear Dr. Terao,

Thank you for submitting your manuscript to PLOS ONE. After careful consideration, we feel that it has merit but does not fully meet PLOS ONE’s publication criteria as it currently stands. Therefore, we invite you to submit a revised version of the manuscript that addresses the points raised during the review process.

We look forward to receiving your revised manuscript.

Kind regards,

Geelsu Hwang, Ph.D.

Academic Editor

PLOS ONE

Journal Requirements:

4. Please amend your authorship list in your manuscript file to include author Dr. Satoru Hirayama.

Reviewers' comments:

Reviewer's Responses to Questions

**Comments to the Author**

1. Is the manuscript technically sound, and do the data support the conclusions?

Reviewer #1: Yes

Reviewer #2: Partly

2. Has the statistical analysis been performed appropriately and rigorously? 

Reviewer #1: Yes

Reviewer #2: Yes

3. Have the authors made all data underlying the findings in their manuscript fully available?

Reviewer #1: Yes

Reviewer #2: Yes

4. Is the manuscript presented in an intelligible fashion and written in standard English?

Reviewer #1: Yes

Reviewer #2: Yes

5. Review Comments to the Author

Reviewer #1: Takizawa et al. used OUFBW to examine the sterilization effect against bacteria that inhabit the oral cavity and some bacteria that are a problem in hospital infections. And it was clarified that OUFBW accounts for the bactericidal effect against those bacteria. We also found that OUFBW has an effective bactericidal effect on toothbrushes and gauze contaminated with these bacteria, and that OUFBW has no harmful effect on gingival cells.

The above results suggest that OUFBW may play an effective role in hygiene management and oral cleaning in the oral area, and are considered to be very important findings.

After reviewing this manuscript, I note the following concerns.

After adjusting OUFBW to various concentrations, we are investigating the bactericidal effect in a state where 100 times the amount of the culture solution containing bacteria is added. In that regard, I would like to ask the following questions.

1) There is no specific description of how to dilute OUFBW in this case. I would like you to describe it.

2) Is there any change in the effect of OUFBW compared to the OUFBW dilution method (dilution using a buffer solution) used to investigate the harmful effects of this dilution method on gingival cells? I think that this point should be clarified.

3) In 2), if unbuffered OUFBW is used for bacterial sterilization and it contains a higher amount of ozone than diluted buffered OUFBW. I think that the sterilization effect of bacteria should be performed.

Reviewer #2: Terao et al. reported an assessment of using OUFBW as a disinfectant against oral pathogens. While the study results show a promising antimicrobial effect from OUFBW, my concerns are how much pathogens are present in the mixed solution of the OUFBW treatment. Line 104-107 indicated that 1ul bacteria of OD 01-0.5 was mixed with 1ml OUFBW, which is 1000 time dilution. OD 0.1-0.5 is also a wide range; how much bacteria in terms of CFU were used as a baseline? Figures 2 and 4 only reported the percentage of pathogens; reporting the actual CFU for each condition is needed to understand the potent and limit of the OUFBW comprehensively.

6. PLOS authors have the option to publish the peer review history of their article (what does this mean?). If published, this will include your full peer review and any attached files.

Reviewer #1: No

Reviewer #2: No

---

## [Author Response · Author response to Decision Letter 0]

15 Mar 2023

Response to Editor and Reviewers

Response to Editor 

We thank the editor for the critical suggestions that have helped us improve our manuscript. As indicated in the responses below, we have taken into consideration all these comments and suggestions and addressed each one of them during our revision of the manuscript. 

<Comment #1> Please ensure that your manuscript meets PLOS ONE's style requirements, including those for file naming.

<Response> In response to the editor’s suggestion, we ensured that our manuscript and figures complied with PLoS One's style guidlines and modified the file name. 

<Comment #2> We note that the grant information you provided in the ‘Funding Information’ and ‘Financial Disclosure’ sections do not match.

<Response> At the time of resubmission, the ‘Funding Information’ and ‘Financial Disclosure’ have been revised to match their descriptions.

<Comment #3> In your Data Availability statement, you have not specified where the minimal data set underlying the results described in your manuscript can be found.

<Response> Figure 1, 2, 4, and 5 (in the revised version): According to the editor’s suggestion, we modified Figure 1, 2，4, and 5 to comply with PLoS One's data policy. Therefore, it became possible to read the values behind the means from these graphs.

<Comment #4> Please amend your authorship list in your manuscript file to include author Dr. Satoru Hirayama.

<Response> The author list has been updated after we discovered some errors in it. We apologize for this error in the authors’ electronic registration. We removed Dr. Satoru Hirayama from the author list and added Dr. Tomoki Maekawa and Prof. Koichi Tabeta (line 6 in the revised version). Accordingly, we submitted the Request for Change to Authorship and updated the ‘Funding Information’ and ‘Financial Disclosure’. 

Lines 329–330 (in the revised version): According to the change in author list, we updated the acknowledgement section as follows: “We thank Dr. Satoru Hirayama, Dr. Toshihito Isono, Dr. Karin Sasagawa, Dr. Rui Saito, Dr. Yoshihito Yasui (Niigata University), Mr. Tadashi Hiwatashi (Futech-Niigata LLC), and Mr. Koichi Seto (IWASE Company Limited, Niigata, Japan) for their technical support.”

Response to Reviewer 1

We are grateful to Reviewer 1 for their critical comments and suggestions that have helped us improve our paper considerably. In response to your comment, we have included a supporting figure. As indicated in the following responses, we have considered all these comments and suggestions in the revised version of our paper.

<Comment #1> There is no specific description of how to dilute OUFBW in this study.

<Response> Lines 108–109 (in the revised version): According to the reviewer’s suggestion, we have added a description of the dilution method of OUFBW in the bactericidal assay as follows: “To prepare OUFBW containing various ozone concentrations, OUFBW containing 4–6 ppm ozone was serially diluted with distilled water.”.

<Comment #2> Is there any change in the effect of OUFBW compared to the OUFBW dilution method (dilution using a buffer solution) used to investigate the harmful effects of this dilution method on gingival cells?　

<Comment #3> In 2), if unbuffered OUFBW is used for bacterial sterilization and it contains a higher amount of ozone than diluted buffered OUFBW. I think that the sterilization effect of bacteria should be performed.

<Response to comments #2 and #3> Lines 151–153 and S1 figure (in the revised version): To answer the reviewer’s queries, we added a description of the preparation of OUFB-PBS in the cytotoxicity assay in lines 151–153 as follows: “For cytotoxicity assay, we prepared OUFB-phosphate-buffered saline (PBS) by diluting OUFBW with 10 × PBS (NACALAI TESQUE INC, Kyoto, Japan).” The ozone concentrations in OUFBW and OUFB-PBS used in our experiments were 1–5 ppm, as shown Figures 2, 5, and S1. 

We agree with the reviewer’s comments, and we performed additional experiments about bactericidal activity of OUFB-PBS three times. Thereby the new finding was brought. We are grateful for reviewer’s critical comments. We have added descriptions about bactericidal activity of OUFB-PBS to results and discussion section and supporting information to the revised manuscript as follows: “OUFB-PBS exerts bactericidal effects against gram-positive and negative bacteria. We conducted additional experiment to investigate the bactericidal activity of OUFB-PBS. For this experiment, we used S. pneumoniae strain D39 as a Gram-positive bacteria and P. aeruginosa strain RIMD 1603003 as a Gram-negative bacteria. These bacteria were exposed to OUFB-PBS containing various ozone concentrations (0.25–5 ppm) for 1 min. S1 Fig shows that exposure to ≥ 2 ppm OUFB-PBS resulted in a > 99% decrease in S. pneumoniae, and exposure to ≥ 1 ppm OUFB-PBS resulted in a > 99% decrease in P. aeruginosa. These findings suggest that OUFB-PBS also exerts bactericidal effects.” In lines 255–263, and “In addition, fig 2 and S1 showed that there was slight difference between OUFBW and OUFB-PBS on the bactericidal activity. The effects of solvents on ozone and nanobubbles have not been investigated, therefore we supposed it should be addressed in future work.” in lines 288–291, and “S1 Fig. Bactericidal effects of OUFB-PBS against S. pneumoniae strain D39 and P. aeruginosa strain RIMD 1603003. S. pneumoniae D39 and P. aeruginosa strain RIMD 1603003 were exposed to 0.4–5 ppm OUFBW for 1 min. Data are presented as the mean ± SD of triplicate experiments and were evaluated using analysis of variance with Dunnett’s multiple-comparisons test. *P < 0.05 compared to the control group. ND stands for undetected and indicates below the detection limit (< 100 CFU/mL).” in lines 452–458.

Response to Reviewer 2

We thank Reviewer 2 for the critical comments and suggestions that have helped us improve our paper considerably. As indicated in the following responses, we have considered the comments and suggestions in the revised version of our article.

<Comment> Figures 2 and 4 only reported the percentage of pathogens; reporting the actual CFU for each condition is needed to understand the potent and limit of the OUFBW comprehensively.

<Response> Table 1, Figure 2 and 4, and lines 191–195 (in the revised version): According to the reviewer’s suggestion, we modified Table 1 (the revised manuscript version on page 11) and, Figures 2 and 4 to read the values of CFU/mL of bacterial solution used in each experiment. Accordingly, we have modified the manuscript as follows: “Table 1. The number of colony-forming units of various bacterial species exposed to ozone ultrafine bubble water. The number of colony-forming units (CFU) of various bacterial species was measured after exposing to ozone ultrafine bubble water containing 4–6 ppm of ozone for 10–300 s. ND stands for undetected and indicates below the detection limit (< 100 CFU/mL).” in lines 191–195.

Lines 199, 200, 238, 239, 253, and 254 (in the revised version): We modified our manuscript to include the description of the detection limit.

Lines 221 and 229 (in the revised version): We modified our manuscript as follows: “Contaminated toothbrushes were immersed in approximately 5 ppm OUFBW or distilled water for 5 min. Exposure to OUFBW resulted in a > 90% decrease in the bacterial load of S. pneumoniae compared to that of the distilled water treatment group (Fig 4a).” in line 221, “Contaminated gauzes were immersed in approximately 5 ppm OUFBW or distilled water for 5 min. Exposure to OUFBW resulted in a > 90% decrease in the bacterial load of P. aeruginosa compared to that of the distilled water treatment group (Fig 4b).” in line 229.

//

---

## [Decision Letter · Decision Letter 1]

23 Mar 2023

Ozone ultrafine bubble water exhibits bactericidal activity against pathogenic bacteria in the oral cavity and upper airway and disinfects contaminated healthcare equipment

PONE-D-23-01776R1

Dear Dr. Terao,

We’re pleased to inform you that your manuscript has been judged scientifically suitable for publication and will be formally accepted for publication once it meets all outstanding technical requirements.

Kind regards,

Geelsu Hwang, Ph.D.

Academic Editor

PLOS ONE

Additional Editor Comments (optional):

Reviewers' comments:

Reviewer's Responses to Questions

**Comments to the Author**

1. If the authors have adequately addressed your comments raised in a previous round of review and you feel that this manuscript is now acceptable for publication, you may indicate that here to bypass the “Comments to the Author” section, enter your conflict of interest statement in the “Confidential to Editor” section, and submit your "Accept" recommendation.

Reviewer #1: All comments have been addressed

Reviewer #2: All comments have been addressed

2. Is the manuscript technically sound, and do the data support the conclusions?

Reviewer #1: Yes

Reviewer #2: Yes

3. Has the statistical analysis been performed appropriately and rigorously? 

Reviewer #1: Yes

Reviewer #2: Yes

4. Have the authors made all data underlying the findings in their manuscript fully available?

Reviewer #1: Yes

Reviewer #2: Yes

5. Is the manuscript presented in an intelligible fashion and written in standard English?

Reviewer #1: Yes

Reviewer #2: Yes

6. Review Comments to the Author

Reviewer #1: The authors responded appropriately to questions from the reviewers and rated the quality of the manuscript as acceptable.

Reviewer #2: My comments are addressed. I don't have additional comments.

7. PLOS authors have the option to publish the peer review history of their article (what does this mean?). If published, this will include your full peer review and any attached files.

Reviewer #1: No

Reviewer #2: No

---

## [Editor Report · Acceptance letter]

3 Apr 2023

PONE-D-23-01776R1 

Ozone ultrafine bubble water exhibits bactericidal activity against pathogenic bacteria in the oral cavity and upper airway and disinfects contaminated healthcare equipment 

Dear Dr. Terao:

I'm pleased to inform you that your manuscript has been deemed suitable for publication in PLOS ONE. Congratulations! Your manuscript is now with our production department. 

Kind regards, 

on behalf of

Dr. Geelsu Hwang 

Academic Editor

PLOS ONE